# Assembling Ultrafine SnO_2_ Nanoparticles on MIL-101(Cr) Octahedrons for Efficient Fuel Photocatalytic Denitrification

**DOI:** 10.3390/molecules26247566

**Published:** 2021-12-14

**Authors:** Ruowen Liang, Shihui Wang, Yi Lu, Guiyang Yan, Zhoujun He, Yuzhou Xia, Zhiyu Liang, Ling Wu

**Affiliations:** 1Province University Key Laboratory of Green Energy and Environment Catalysis, Ningde Normal University, Ningde 352100, China; t1629@ndnu.edu.cn (R.L.); wangshihui21@163.com (S.W.); luyi080391@163.com (Y.L.); ygyfjnu@163.com (G.Y.); hhzzjj1990@163.com (Z.H.); 2Fujian Provincial Key Laboratory of Featured Materials in Biochemical Industry, Ningde Normal University, Ningde 352100, China; 3State Key Laboratory of Photocatalysis on Energy and Environment, Fuzhou University, Fuzhou 350002, China; 4Xiamen Ocean Vocational College, Xiamen 361000, China

**Keywords:** MIL-101(Cr), SnO_2_, photocatalysis, pyridine, denitrification

## Abstract

Effectively reducing the concentration of nitrogen-containing compounds (NCCs) remains a significant but challenging task in environmental restoration. In this work, a novel step-scheme (S-scheme) SnO_2_@MCr heterojunction was successfully fabricated via a facile hydrothermal method. At this heterojunction, MIL-101(Cr) octahedrons are decorated with highly dispersed SnO_2_ quantum dots (QDs, approximate size 3 nm). The QDs are evenly wrapped around the MIL-101(Cr), forming an intriguing zero-dimensional/three-dimensional (0D/3D) S-scheme heterostructure. Under simulated sunlight irradiation (280 nm < *λ* < 980 nm), SnO_2_@MCr demonstrated superior photoactivity toward the denitrification of pyridine, a typical NCC. The adsorption capacity and adsorption site of SnO2@MCr were also investigated. Tests using 20%SnO_2_@MCr exhibited much higher activity than that of pure SnO_2_ and MIL-101(Cr); the reduction ratio of Cr(VI) is rapidly increased to 95% after sunlight irradiation for 4 h. The improvement in the photocatalytic activity is attributed to (i) the high dispersion of SnO_2_ QDs, (ii) the binding of the rich adsorption sites with pyridine molecules, and (iii) the formation of the S-scheme heterojunction between SnO_2_ and MIL-101(Cr). Finally, the photocatalytic mechanism of pyridine was elucidated, and the possible intermediate products and degradation pathways were discussed.

## 1. Introduction

Liquid fuels are widely used in electricity and transportation systems. Crude gasoline fuel naturally contains high concentrations of nitrogen-containing compounds (NCCs) such as pyridine, indoles, nitrides, and their derivatives. The combustion products (e.g., NO_x_ and unburned hydrocarbon particles) are released into the environment, causing photochemical smog and exerting serious hazardous effects on ecosystems and human health [1,2]. As environmental regulations become more stringent, the removal of NCCs from fuel has become increasingly more important. To meet the quality demands of liquid fuels with low nitrogen content, the nitrogen species can be removed by catalytic hydrodenitrification (HDN). In fact, HDN is regarded as the classic technology of nitrogen-species removal from fuels [3] but requires high temperatures, high pressures, and very active catalysts. Current research has focused on photocatalytic one-pot removal of NCCs, mainly because the involved reactions are “green”, mild, and directly utilize sunlight. Thus far, only a few semiconductors such as Bi_20_TiO_32_, TiO_2_/Fe_2_O_3_, and Bi_2_MoO_6_/CdS have been explored as fuel denitrification photocatalysts [4,5,6,7]. Exploring more effective photocatalysts for gasoline fuel denitrification remains among the most attractive topics in fuel purification research.

Metal-organic frameworks (MOFs), a class of crystalline porous materials with tailored pores/structures and high surface areas, have attracted great interest in various fields [8]. Owing to their high specific surface area, adjustable pore size, and easily modified surfaces, MOFs are strong candidates for photocatalysis [9,10,11,12]. The application of MOFs as photocatalysts was reported as early as 2004 by Garcia et al. [13]. Many MOFs, for example, MIL-125(Ti)–NH_2_, MIL-68(In)–NH_2_, MIL-53(Fe), and MIL-101(Cr), possess the semiconductor characteristics required of photocatalysts [14,15,16,17]. However, the deficient active sites and low separation efficiency of the photoinduced carriers reduce the photocatalytic performance of pure MOFs [18,19]. To further improve their photoactivities, many researchers proposed to construct MOFs-based heterostructures, such as Type-II heterojunction, Z-scheme heterojunction, and step-scheme (S-scheme) heterojunction [20,21,22,23]. Our group has also designed a novel sandwich-like hierarchical AgBr–Ag@MIL-68(Fe) S-scheme photocatalyst with excellent photocatalytic activity for Cr(VI) reduction and dye degradation [11].

SnO_2_, a chemically stable n-type semiconductor with an approximate bandgap energy of 4.0 eV, has attracted much interest for its superior photocatalytic oxidation performance. This high performance is attributable to the valence band (VB) edge of SnO_2_, which is located at ca. 3.9 eV vs. NHE [24,25]. However, the slow electron transfer dynamics and weak reduction potentials of the conduction band (CB) electrons (with energy levels of ca. −0.1 eV vs. NHE) restrict the applications of SnO_2_ in photocatalytic reactions. S-scheme coupling promises to overcome the drawbacks of SnO_2_ and MOFs by deliberately sacrificing the holes and electrons with poor redox ability and retaining those with strong redox ability. However, the existing reports on SnO_2_@MOFs have focused only on lithium-ion batteries or electrochemical sensing [26,27]; reports on SnO_2_@MOFs for photocatalytic applications are rather scarce. MOF-based S-scheme photocatalysts for fuel denitrification are unavailable at present.

MIL-101(Cr) (Cr_3_F(H_2_O)_2_O[(O_2_C)C_6_H_4_(CO_2_)]_3_.nH_2_O, *n* ≈ 25) is a Cr-based MOF synthesized by G. Férey et al. in 2005 [28]. In previous work, it was chosen as the target supporter owing to its high chemical stability, water stability, and photoresponsiveness [29,30,31,32]. Inspired by the unique features of MIL-101(Cr), we designed a zero-dimensional/three-dimensional (0D/3D) S-scheme heterojunction involving MIL-101(Cr) octahedrons decorated with SnO_2_ quantum dots (QDs) (denoted by SnO_2_@MCr). The first fabrication step prepares the MIL-101(Cr) octahedrons. Next, SnO_2_ quantum dots (QDs or nanoparticles) are anchored on the rhombic octahedral MIL-101(Cr) crystals via a facile in situ hydrothermal strategy. The photocatalytic activities of the as-prepared SnO_2_@MCr samples are investigated in the denitrification of pyridine, a typical NCC, under simulated sunlight irradiation (280 nm < *λ* < 980 nm). The decomposition process of pyridine and the formation of intermediate products are systematically investigated, and the surface microstructure of the photocatalyst is related to its photocatalytic performance. Finally, the possible mechanism of the decomposition process is deduced.

## 2. Results

### 2.1. Characterizations

A series of SnO_2_@MCr composites was prepared via a two-step synthetic procedure (see Figure 1). Figure 1a shows the XRD patterns of the as-prepared samples. All diffraction peaks of the MIL-101(Cr) sample were well matched to the calculated peaks, indicating the successful synthesis of MIL-101(Cr) materials [28,30]. The diffraction peaks at 2*θ* = 27.14°, 34.87°, and 52.74° in Figure 1b were consistent with the diffractions of the (110), (101), and (211) planes of tetragonal SnO_2_, respectively (JCPDS no. 00-021-1250) [24,25]. Meanwhile, all characteristic diffraction peaks of SnO_2_ and MIL-101(Cr) were found in the SnO_2_@MCr composites, indicating the successful introduction of SnO_2_ on the MIL-101(Cr) architecture. The diffraction intensity of MIL-101(Cr) in the nanocomposite gradually weakened from that of pure MIL-101(Cr) as the loading amount of SnO_2_ QDs increased, suggesting a covering effect. Moreover, the characteristic peaks of the SnO_2_ QDs were rather flatter than those of pure SnO_2_. Using the Debye–Scherrer equation [33], this flattening can be ascribed to the small size and highly uniform distribution of the SnO_2_ QDs on the MIL-101(Cr) surfaces. In an inductively coupled plasma–atomic emission spectrometry analysis, the amounts of SnO_2_ doped on the MIL-101(Cr) surface were confirmed as 4.33, 9.21, 17.64, 27.74, and 37.65 wt.%, slightly lower than their theoretical values of 5, 10, 20, 30, and 40 wt.%, respectively.

Appendix A shows the Fourier transform infrared (FTIR) spectra of the samples. In the spectra of MIL-101(Cr), the peaks at 1402, 1622, and 1709 cm^−1^ were attributed, respectively, to the aromatic O=C=O, C–C, and C–O stretching vibration modes of the H_2_BDC ligand, respectively [34]. The characteristic peak pair in the 800–500 cm^−1^ range typified the absorption peaks of Cr–O. In the spectrum of SnO_2_, the bands between 800 and 500 cm^−1^ were ascribed to the stretching vibrations of Sn–O [24]. Since SnO_2_ contains no special functional groups on its surface, the FTIR spectrum of 20%SnO_2_@MCr was similar to that of MIL-101(Cr), further implying that the MIL-101 (Cr) structure was maintained after the hydrothermal process.

The morphologies of the samples were investigated by scanning electron microscopy (SEM). Pure MIL-101(Cr) exhibited a diagnostic octahedral structure and smooth surface (Figure 2a), whereas SnO_2_ appeared as irregular nanoparticles (Figure 2b). The particles of SnO_2_@MCr were shaped similarly to those of MIL-101(Cr), but the smooth surfaces of the MIL-101(Cr) octahedrons were roughened after introducing the SnO_2_ (Figure 2c,d), because they were distinctly covered by uniform SnO_2_ clusters. The microstructures of the SnO_2_@MCr composites were further investigated through transmission electron microscopy (TEM) and high-resolution TEM (HRTEM) analyses. The pure MIL-101(Cr) sample exhibited an octahedral microstructure with an approximate particle diameter of 600 nm (Figure 3a,b). Pure SnO_2_ presented a quasi-spherical morphology consisting of agglomerated particles with a mean diameter of 5–6 nm (Figure 3c–f). 

Taking 5%SnO_2_@MCr composite as an example, the SnO_2_ particles, when grown on MIL-101(Cr), were highly dispersed, and the SnO_2_ QDs were tightly wrapped around the MIL-101(Cr) surface (Figure 4a–c). Note that the octahedral morphology of MIL-101(Cr) was maintained after growing SnO_2_ in situ. In the representative HRTEM image of SnO_2_ (Figure 4d), the lattice fringes were separated by 0.34 nm. This spacing corresponds to the interplanar distance of the (110) planes of tetragonal SnO_2_. Unlike pure SnO_2_, the SnO_2_ QDs in the SnO_2_@MCr composites with approximate diameters of 3–5 nm were highly dispersed on the MIL-101(Cr) surface (Figure 4e). No agglomerates or isolated SnO_2_ clusters were detected in the nanocomposites. This uniformity can be attributed to the high specific surface area of MIL-101(Cr), enabling control of the morphology and particle size of SnO_2_. As for the samples of 10%SnO_2_@MCr and 20%SnO_2_@MCr, increasing the amount of SnO_2_ increased the coverage of SnO_2_ QDs on the MIL-101(Cr) surface and reduced the size of the SnO_2_ QDs (Figure 4f–o), which may be attributed to the augmentation of the interface contact area between MIL-101(Cr) and SnO_2_. However, an aggregating phenomenon appeared when the SnO_2_ amount was increased to 30 wt.% (Figure 4p–y).

The spatial distributions and elemental compositions of the samples were analyzed by energy-dispersive X-ray spectroscopy (EDS). The full EDS spectrum of 20%SnO_2_@MCr is shown in Appendix A. The spectrum presents the signal peaks of C, Cr, O, and Sn, confirming the existence of the prepared MIL-101(Cr) and SnO_2_ in the composite samples. As further evidence, the EDS mapping images (Appendix A–d) displayed the homogeneous distribution of C, O, Cr, and Sn elements through the composite, illustrating that SnO_2_ was successfully loaded on MIL-101(Cr) to form a heterogeneous structure.

The surface compositions and chemical states of the samples were elucidated by X-ray photoelectron spectroscopy (XPS). The survey XPS spectrum (Figure 5a) of SnO_2_@MCr was dominated by C, O, Cr, and Sn, consistent with the EDS results. Meanwhile, the C 1s spectrum of MIL-101(Cr) showed three peaks at 284.46, 285.34, and 288.57 eV, corresponding to C–C/C=C, C–O, and C=O of H_2_BDC, respectively [34,35] (Figure 5b). The two peaks at 587.16 and 577.69 eV in Figure 5c were attributed to the 2p_1/2_ and 2p_3/2_ signals of Cr, respectively, demonstrating the presence of Cr^3+^ in MIL-101(Cr) [35]. Importantly, the dominant peaks of C 1s and Cr 2p in the spectrum of the SnO_2_@MCr sample shifted to higher binding energies from those of pure MIL-101(Cr). The corresponding binding energies are summarized in Table 1. In the high-resolution Sn 3d XPS spectrum of pure SnO_2_ (Figure 5d), the binding energies at 487.36 and 495.73 eV were assigned to 3d_5/2_ and 3d_3/2_ of Sn, respectively, indicating the presence of Sn^4+^. In the Sn 3d spectrum of the SnO_2_@MCr sample, the Sn 3d_5/2_ and Sn 3d_3/2_ peaks shifted to 487.19 and 496.19 eV, respectively, from those of SnO_2_. This shift was accompanied by a lower-energy shift of the Cr 2p and C 1s peaks, indicating a decreased electron density on the Cr and C moieties and an increased electron density on the Sn moieties, while a built-in electric field formed at the SnO_2_@MCr interface [36]. Collectively, these results indicate a strong interaction between SnO_2_ and MIL-101(Cr) in SnO_2_@MCr rather than a simple physical mixture. All XPS results implied that SnO_2_ QDs were successfully loaded on the MIL-101(Cr) surface, thus fabricating the S-scheme SnO_2_@MCr photocatalytic system.

The Brunauer–Emmett–Teller (BET) surface areas and pore structures of the prepared samples were determined from adsorption–desorption measurements and are displayed in Figure 6a,b. The nitrogen adsorption–desorption isotherm of the original MIL-101(Cr) was a type I isotherm on the IUPAC classification scheme. Together with the corresponding pore-size distribution, this result indicates a mesoporous structure [34] (Figure 6b). Table 2 lists the surface areas and pore volumes of the samples. The BET surface area and pore volume of MIL-101(Cr) were 3834 m^2^/g and 1.66 cm^3^/g, respectively. When the SnO_2_ QDs were introduced, they occupied the pores of MIL-101(Cr), so the BET surface area and pore volume decreased. However, they still exceeded those of the SnO_2_ sample (pore volume = 161 m^2^/g; see Appendix A).

Figure 7a shows the optical absorption spectra of all samples. The spectrum of pure SnO_2_ displayed an absorption edge around 370 nm. The spectrum of pristine MIL-101(Cr) exhibited intense absorption in both the UV and visible regions, with two characteristic peaks. The absorption band of MIL-101(Cr) in the UV region was assigned to π-π* transitions of the ligands and Cr (III)–oxide clusters, whereas the visible-light absorption came from Cr^3+^–oxide clusters, which characteristically absorbed at 450 and 600 nm (as also reported in the literature [17,34,37]). In the spectra of the SnO_2_@MCr samples, the absorption edge of SnO_2_ was undisturbed after the SnO_2_ QDs was grown on the on MIL-101(Cr), possibly because the SnO_2_ QDs could not alter the crystal lattice of MIL-101(Cr) [38]. The Mott–Schottky plots analyzed at 500 and 1000 Hz are displayed in panels (b) and (c) of Figure 7, respectively. The flat band potentials (*V*_fb_) of MIL-101(Cr) and SnO_2_ were −0.95 and −0.25 eV vs. the Ag/AgCl electrode, respectively. Transforming the difference between the Ag/AgCl electrode and the standard hydrogen electrode, the CB positions of the samples were finally calculated as −0.75 and −0.05 eV, respectively (vs. NHE, pH = 7). 

The VBs of the samples were measured from the VB–XPS plots. Panels (d) and (e) of Figure 7 display the VB-XPS spectra of MIL-101(Cr) and SnO_2_, respectively, from which the VBs were determined as 1.77 and 4.05 eV, respectively. The VB potential of the normal hydrogen electrode (E_VB-NHE_, vs. NHE, pH = 7) was calculated as the contact potential difference between the sample and the XPS analyzer, namely, as E_VB-NHE_ = φ + E_VB-XPS_–4.44 [23]. Here, φ is the electron work function (4.55 eV) of the XPS analyzer, and E_VB-XPS_ is the VB measured from the VB–XPS plots. By this equation, the E_VB-NHE_ values of MIL-101(Cr) and SnO_2_ were calculated as 1.88 and 4.16 eV, respectively. From the Mott–Schottky and XPS spectral analyses, the calculated bandgaps of MIL-101(Cr) and SnO_2_ were 2.63 and 4.21 eV, respectively.

To further explore the interfacial charge transfer between MIL-101(Cr) and SnO_2_, the work functions of MIL-101(Cr), SnO_2_, and SnO_2_@MCr were measured and calculated from the VB–XPS plots. When the solid sample made good electrical contact with the metal sample holder of the XPS analyzer and the electron transfer was balanced, the Fermi levels of the solid sample and metal holder reached the same level. However, as the work functions differed between the two materials, the kinetic energy of the free electrons was changed by the contact potential difference ΔV = Φ − φ (where φ = 4.55 eV is the work function of the XPS analyzer, and Φ is the work function of the sample), thereby changing the binding energy of the electrons [39]. After measuring the binding energy changes over a small range by XPS, ΔV was obtained from the spacing between the two inflection points of the curves (Figure 7d–f). The Φs of MIL-101(Cr), SnO_2_, and SnO_2_@MCr were thus calculated as 5.53, 6.02, and 5.76 eV, respectively. Figure 8 displays the band structures and work functions of MIL-101(Cr) and SnO_2_ before and after contact, derived from the above-obtained Mott–Schottky and VB XPS values. MIL-101(Cr) was identified as a reduction-type photocatalyst with a lower work function (5.53 eV) and a higher Fermi level (*E*_f_), whereas SnO_2_ was an oxidation-type photocatalyst with a higher work function (6.02 eV) and lower *E*_f_. When MIL-101(Cr) and SnO_2_ were in contact, electrons were transferred from MIL-101(Cr) to SnO_2_ until the *E*_f_ values were equalized. At this time, an internal electric field from MIL-101(Cr) to SnO_2_ formed at the interface. Once irradiated with light, the electrons in the CB of SnO_2_ recombined with the holes in the VB of MIL-101(Cr); meanwhile, the holes of SnO_2_ with a strong oxidation capacity and the photogenerated electrons of MIL-101(Cr) with a strong reduction capacity remained in their energy bands, greatly promoting the redox ability of the S-scheme SnO_2_@MCr photocatalyst.

### 2.2. Adsorption Performance

Substrate adsorption and mass transmission are essential in the photocatalytic process. Therefore, pyridine adsorption experiments were carried out using MIL-101(Cr), SnO_2_, and 20%SnO_2_@MCr as adsorbents. The SnO_2_ nanoparticles were almost unable to adsorb pyridine, whereas the MIL-101(Cr) exhibited a 38.8% removal efficiency for pyridine (Appendix A). After being coated with SnO_2_ QDs, the removal efficiency for pyridine decreased to 15.4%, which can be ascribed to acid–base interactions (chemical adsorption) and pore-adsorption interactions (physical adsorption) between MIL-101(Cr) and the pyridine molecules.

To further understand the interaction mechanisms between the representative organic molecules and the SnO_2_@MCr surface, high-resolution Cr and Sn spectra were obtained (presented in Figure 9). The Cr 2p bands of the bare SnO_2_@MCr surface at 577.75 and 587.26 eV were ascribed to Cr 2p_3/2_ and Cr 2p_1/2_, respectively (Figure 9a). After treating the MIL-101(Cr) surface with pyridine, the Cr 2p_3/2_ and Cr 2p_1/2_ bands shifted to 577.23 and 586.95 eV, respectively. Meanwhile, pyridine absorption did not significantly influence the Sn 3d_5/2_ and Sn 3d_3/2_ peaks in the spectrum of SnO_2_@MCr (Figure 9b). These results prove that pyridine strongly adsorbed on the SnO_2_@MCr surface and the main binding sites were derived from Cr^3+^. 

To determine the acidities of the samples, pyridine adsorption was monitored by FTIR (FTIR-pyridine). In the FTIR-pyridine spectrum of 20%SnO_2_@MCr, the sharp peaks at 1450 cm^−1^ indicate the presence of Lewis-acidic uncoordinated octahedral Cr^3+^ sites (Figure 9c). The peak at 1540 cm^−1^ was ascribed to Brønsted acid sites contributed to by uncoordinated water molecules of MIL-101(Cr) (forming protonated pyridine, PyH^+^) [40]. The concentrations of the Lewis and Brønsted acid sites in 20%SnO_2_@MCr were 108.7 and 15.1 µmol/g, respectively, suggesting a mixed acidity in 20%SnO_2_@MCr with dominant Lewis acidity. Moreover, the Cr^3+^ central metal in SnO_2_@MCr was predicted as a Lewis acid, which can combine with pyridine (a Lewis base). Therefore, Cr and N might interact during the pyridine adsorption process over SnO_2_@MCr. The above analysis showed the strong pyridine binding affinity of MIL-101(Cr), while the synergistic effect of MIL-101(Cr) and SnO_2_ ensured the high-efficiency denitrification of pyridine over the SnO_2_@MCr nanocomposite.

### 2.3. Photocatalytic Properties

The photocatalytic activity of SnO_2_@MCr was evaluated during photocatalytic denitrification of pyridine under simulated sunlight irradiation. First, the photocatalytic nature of the reaction was demonstrated in blank experiments (Figure 10a). The pyridine denitrogenation hardly occurred in the absence of light or photocatalyst. The denitrogenation abilities of pristine SnO_2_ and MIL-101(Cr) were relatively weak, and the pyridine concentration changes were small. In contrast, the pyridine denitrogenation proceeded smoothly in the presence of SnO_2_@MCr. In the sample containing 20 wt.% SnO_2_, the photo-denitrification efficiency rapidly increased (to 90.1%) after four hours of simulated sunlight irradiation. To check the synergetic effect, the photocatalytic activity of SnO_2_ + MIL-101(Cr) (prepared by simply mixing SnO_2_ and MIL-101(Cr) in the proper proportions) was studied under the same condition. In this mixture, 65.0% of the pyridine was denitrogenated after four hours of sunlight irradiation, further confirming that the SnO_2_@MCr heterojunction supported carrier transfer between the interface of SnO_2_ and MIL-101(Cr), thus improving the photocatalytic degradation efficiency. However, increasing the SnO_2_ content beyond 20 wt.% reduced the photocatalytic performance. As evidenced in Figure 10b, the denitrification efficiency for pyridine decreased from 90.1% to 61.0%. This result is reasonable, because an excessively high SnO_2_ content not only facilitates agglomeration of the SnO_2_ particles, but also shields the active sites on MIL-101(Cr) [30]. To understand the reaction kinetics of the photocatalytic pyridine denitrification, the reaction rate constant *k* was calculated from the expression ln(*C_t_*/*C*_0_) = −*k_t_*, which assumes a pseudo first-order reaction. Here, *C*_0_ and *C_t_* denote the pyridine concentrations at times 0 and *t*, respectively. Under simulated sunlight irradiation, the 20%SnO_2_@MCr photocatalyst achieved a pyridine denitrification *k* of 0.5012 h^−1^, higher than those of MIL-101(Cr) (0.1888 h^−1^), SnO_2_ (0.2275 h^−1^), and SnO_2_+MIL-101(Cr) (0.2229 h^−1^) (Figure 10c,d). A comparison between the photocatalytic activity of our 20%SnO_2_@MCr and that of other reported catalysts is listed in Table 3. It is worth noting that, compared with some of photocatalysts, the 20%SnO_2_@MCr exhibits better or comparable photocatalytic activity for the pyridine denitrogenation under visible light irradiation [2,4,5,6,7,41,42].

Reusability is another important property of a practical photocatalyst. The 20%SnO_2_@MCr photocatalyst delivered a stable performance with no noticeable deactivation over five cycles of photocatalytic experiments (Figure 11a). Powder XRD patterns and TEM images confirmed that the composition and morphology of 20%SnO_2_@MCr were well preserved after the photocatalysis (Figure 11b–e), affirming the excellent long-term durability of the photocatalyst under simulated sunlight irradiation. The intermediate products of pyridine denitrification were then determined by high-performance liquid chromatography (HPLC)–MS analysis (see Appendix A). Under irradiation for four hours, the peak intensity of pyridine (m/z ~80) was greatly decreased, implying the successful denitrogenation of pyridine. Meanwhile, two new peaks gradually appeared at m/z = 64 and m/z = 58, suggesting that pyridine was transformed into protonated intermediate products and was ultimately mineralized into inorganic products such as CO^2^ and NO^3-^, again consistent with our previous results [2,7].

### 2.4. Possible Photocatalytic Mechanism 

The generation of various oxidative species over the 20%SnO_2_@MCr photocatalyst was probed by the electron spin resonance (ESR) technique [43,44]. The characteristic peaks of DMPO-•O_2_^−^ and DMPO-•OH were not observed under dark conditions but appeared under light irradiation (Figure 12a,b). Judging from this result, •O_2_^−^ and •OH were the main active species. The generated h^+^ were investigated using 2,2,6,6-tetramethyl-piperidinyl-1-oxyl (TEMPO) as the hole probe because its radical can be oxidized by h^+^. As shown in Figure 12c, the characteristic TEMPO peak was strong under dark conditions and significantly decreased with increasing irradiation time, confirming that holes were photogenerated during pyridine denitrification. The above experiments prove that the main active species in the 20%SnO_2_@MCr S-scheme heterostructure are •O_2_^−^, •OH, and h^+^, further affirming the forming of the S-scheme heterojunction in this SnO_2_@MCr system.

The photogenerated carrier separation ability in 20%SnO_2_@MCr was evaluated in a photocurrent response and electrochemical impedance spectroscopy analysis. Figure 13a displays the transient photocurrent responses of MIL-101(Cr), SnO_2_, and 20%SnO_2_@MCr during several on–off cycles under simulated sunlight irradiation. The current generation implied that electron–hole pairs were separated under the simulated sunlight irradiation, whereas the current decay indicated recombination of the photogenerated carriers. The 20%SnO_2_@MCr sample achieved a clearly higher photocurrent intensity than pure MIL-101(Cr) and SnO_2_, implying that this composite optimally separated and transferred the carriers. Figure 13b shows typical Nyquist plots of the prepared samples obtained in the dark. The smaller radius of 20%SnO_2_@MCr than of the pure substances indicates a small charge-transfer impedance and a high charge separation in the composite. In summary, the 20%SnO_2_@MCr sample achieved the largest photocurrent and the smallest electrical impedance among the prepared samples, indicating that it optimized the photocatalytic performance, as shown in the previous photoactivity test results.

Based on the above results, the possible scheme of pyridine denitrogenation by SnO_2_@MCr was proposed as Figure 2. The band structure suggested that the SnO_2_@MCr heterojunction was a typical II-type heterojunction. However, if SnO_2_@MCr obeyed the II-type mechanism, the electrons of MIL-101(Cr) could be injected into the CB of SnO_2_ (with an energy of −0.03 eV), and the holes of SnO_2_ could migrate to MIL-101(Cr) (with a VB energy of only 1.88 eV). These photogenerated electrons and holes would have lower reducing and oxidizing abilities to generate •O_2_^−^ (O_2_/•O_2_^−^, −0.33 eV) and •OH (OH^−^/•OH, 2.29 eV), which clearly contradicts the above experimental results. Combining the above-discussed energy band structures, internal electric fields, and free radical testing results, the most reasonable photocatalytic charge-transfer mechanism of this experimental system was the S-scheme photocatalytic mechanism. In particular, the formation of a novel S-scheme SnO_2_@MCr heterojunction was facilitated by the large contact area and the Fermi-level difference between SnO_2_ and MIL-101(Cr). The relatively useless electrons (from the CB of SnO_2_) and holes (from the VB of MIL-101(Cr)) recombined, while the useful holes and electrons were retained. The enhanced redox potential at the VB of SnO_2_ and the CB of MIL-101(Cr) enabled the oxidation of H_2_O to •OH and the reduction of O_2_ to •O_2_^−^, respectively. The generated holes, •OH, and •O_2_^−^, were the main active species that reacted with the pyridine molecules adsorbed on the SnO_2_@MCr surface. 

## 3. Materials and Methods

### 3.1. Materials

All reagents were analytical grade and used without further purification. Chromium nitrate (Cr(NO_3_)_3_·9H_2_O, 99 wt%), stannic chloride (SnCl_4_·5H_2_O, 99 wt%), hydrofluoric acid (HF, 49%), anhydrous ethanol (99.5 wt%), pyridine (99.5 wt%), and octane (99.5 wt%) were purchased from Sinopharm Chemical Reagent Co. Ltd., China. Terephthalic acid (H_2_BDC, 99 wt%) was supplied by Alfa Aesar China Co., Ltd. (Tianjin, China).

### 3.2. Synthesis of MIL-101(Cr)

MIL-101(Cr) was synthesized via a modified hydrothermal method reported by G. Férey [28]. Briefly, Cr(NO_3_)_3_·9H_2_O (4.00 g), H_2_BDC (1.64 g), HF (0.1 mL), and deionized water (48 mL) were added to a 100 mL Teflon-lined autoclave, and the mixture was heated at 220 °C for 8 h. After cooling down to room temperature, the product was collected by centrifugation and then further washed with hot ethanol and deionized water to remove the unreacted H_2_BDC. Finally, the obtained pale green powder was dried overnight at 80 °C.

### 3.3. Synthesis of SnO_2_@MIL-101(Cr) (SnO_2_@MCr)

As shown in Figure 1, the SnO_2_@MCr hybrids were synthesized by a simple hydrothermal method. Typically, a certain amount of SnCl_4_·5H_2_O (0.0582, 0.1164, 0.2327, 0.3492, 0.4656 g), 500 mg of MIL-101(Cr) powder, and 50 mL of deionized water were added to a 100 mL Teflon-lined autoclave and the mixture was heated at 160 °C for 10 h. After being cooled to room temperature, the crude product was obtained by centrifugation and then further washed with deionized water and ethanol several times. The obtained pale green powder was dried overnight at 60 °C. The loading of SnO_2_ in the composites was about 5, 10, 20, 30, and 40 wt.% for 0.0582, 0.1164, 0.2327, 0.3492, and 0.4656 g of SnCl_4_·5H_2_O, designated as sample 5%SnO_2_@MCr, 10%SnO_2_@MCr, 20%SnO_2_@MCr, 30%SnO_2_@MCr, and 40%SnO_2_@MCr, respectively. The bare SnO_2_ without adding MIL-101(Cr) was prepared through the same procedure.

### 3.4. Characterizations

XRD patterns were carried on a Bruker D8 Advance X-ray diffractometer operated at 40 kV and 40 mA with Ni-filtered Cu Ka irradiation (λ = 0.15406 nm). The size and the morphology of the samples were determined by scanning electron microscopy (SEM) using a Hitachi SU8000 scanning microscope. Transmission electron microscopy (TEM) and high-resolution transmission electron microscopy (HRTEM) images were obtained using a JEOL model JEM 2010 EX instrument at an accelerating voltage of 200 kV. UV-vis diffuse reflectance spectra (UV-vis DRS) were obtained using a UV-vis spectrophotometer (Varian Cary 500). The Brunauer–Emmett–Teller (BET) surface area was measured with an ASAP2020M apparatus (Micromeritics Instrument Corp., USA). X-ray photo electron spectroscopy (XPS) measurement was performed with a Thermo Scientific ESCA Lab 250 spectrometer. The electron spin response (ESR) experiment was carried out with a Bruker 300 spectrometer. The photoluminescence spectra (PL) of all samples were tested using a fluorescence spectrometer (FLS 980). The photocurrent measurements were conducted with a BAS Epsilon workstation. The Mott–Schottky experiments were conducted on a Precision PARC workstation using electro chemical impedance spectroscopy (EIS). Fourier transform infrared spectroscopy (FTIR) with pyridine as the probe molecule was performed using a Nicolet 6700 to identify the Brønsted and Lewis acid sites. The samples were first degassed at 180 °C for 2 h, followed by pyridine adsorption at 25 °C for 1 h. Removal of physisorbed pyridine was performed at 150 ^°^C for 30 min, and the spectra was collected after conducting desorption at 150 °C for 1 h. The liquid chromatograph-mass spectrometer (HPLC-MS) method for analyzing pyridine was performed using an Agilent 1200 series (Palo Alto, CA, USA) equipped with an Agilent Zorbax Eclipse XDB-C18 column (2.1 mm × 100 mm, 3.5 m). The column was maintained at 30 °C during the sample analysis. The measurement for pyridine was performed in an isocratic elution program with methanol/acetone = 70:30 (*v*/*v*) as the mobile phase. Flow rate was kept at 0.2 mL/min, and the injection volume was 10 μL.

### 3.5. Evaluation of Photocatalytic Activity

The photocatalytic activity of SnO_2_@MCr composites was studied through the photocatalytic denitrification of pyridine under simulated sunlight irradiation. In detail, the photocatalytic denitrification of pyridine was carried out at 30 °C in a 100 mL quartz reactor containing 25 mg of SnO_2_@MCr and 50 mL of pyridine/octane solution (100 μg/g). The simulated NCCs-containing gasoline fuel (100 μg/g) was prepared by dissolving 70 mg of pyridine in 1.0 L of octane. The suspension was stirred in the dark for 2 h to ensure the establishment of adsorption–desorption equilibrium. The suspensions were irradiated by a 300 W Xe lamp (PLS-SXE 300, Beijing Perfectlight Co. Ltd., 280 nm < λ < 980 nm), and the distance between them was 10 cm. During illumination, 2 mL of suspension was taken from the reactor at scheduled intervals and centrifuged to separate the photocatalyst. The pyridine content in the supernatant solution was determined colorimetrically at 251 nm using a Cary 50 UV-vis spectrophotometer (Varian Co.).

## 4. Conclusions

In summary, a novel SnO_2_@MCr photocatalyst was synthesized via a simple hydrothermal strategy. In this architecture, the highly dispersed SnO_2_ QDs with the size of ~3 nm were evenly distributed on the MIL-101(Cr). Moreover, because of the differences of Fermi level and band structure of the SnO_2_ and MIL-101(Cr), an S-scheme heterojunction could be established. The obtained SnO_2_@MCr hybrid exhibited high photocatalytic performances for pyridine denitrification. In particular, after sunlight irradiation for 4 h, 20%SnO_2_@MCr could result in 95% pyridine denitrification, with a corresponding k value of 0.5012 h^−1^. After four cycles of pyridine denitrification, the photocatalytic properties, crystal structure, and microstructure of the composite were maintained, indicating the excellent stability and reusability of SnO_2_@MCr composites. In addition, the possible photocatalytic mechanism of pyridine was elucidated through the systematic characterization results. In addition, it was pointed out that the oxidative radicals, such as holes, •OH and •O_2_^−^ were the main reaction species for pyridine denitrification. In addition, the TiO_2_ amount, initial pH value, catalyst dosage, and initial BPA concentration influenced the degradation of BPA. The improvement in photocatalytic activity is attributed to (i) the high dispersion of SnO_2_ QDs, (ii) the binding of the rich adsorption sites with pyridine molecules. and (iii) the formation of the S-scheme heterojunction between SnO_2_ and MIL-101(Cr). Finally, the synergistic effect of coordination adsorption-photocatalysis, the possible degradation pathways, and the degradation mechanism were proposed. This work not only demonstrates the immense potential of MOFs in photocatalysis but also provides a specimen for further fabrication and design of MOFs-based S-scheme heterojunctions that could be used for effectively removing NCCs in crude gasoline fuel.

## Data Availability

The data presented in the study are available from the corresponding author.

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
