# Peer review of "Assembling Ultrafine SnO2 Nanoparticles on MIL-101(Cr) Octahedrons for Efficient Fuel Photocatalytic Denitrification"

_molecules, 2021, doi:10.3390/molecules26247566_

Round 1

Reviewer 1 Report

The manuscript "Assembling ultrafine SnO2 nanoparticles on MIL-101(Cr) octahedrons for efficient fuel photocatalytic denitrification" shows a new inspired design 0D/3D S-scheme heterojunction involving a Cr-based MOF with SnO2 quantum dots. Authors explore the characterization using several techniques including XRD, FTIR, SEM, TEM (and HRTEM), EDS and XPS. Also, the Adsorption performance and the Photocatalytic properties for the new molecule were shown. The authors suggested a possible photocatalytic mechanism for the molecule. I do not have any suggestions to contribute, so I consider that the article can be published in this form.

Author Response

Thank the reviewer very much for the positive valuation about our work. We hope that the revised manuscript is ready for publication.

Reviewer 2 Report

This article is well written and presents new and interesting researches

Author Response

(The authors gave the same response as above.)

Reviewer 3 Report

In present manuscript authors prepared a heterojunction based on SnO2 and MIL- 27 101(Cr) for the photocatalytic degradation of pyridine. The composites structure showed the better performance owing to improved light harvesting, a high surface area, etc.  They also performed experiments for the reusability and mechanism study. This paper can be recommended for publication after appropriate revising. The main revision suggestions are as follows:

  1. The statement of innovation was not sufficient in the section of Introduction. Introduction should be clearly stated research questions and targets first. Then answer several questions: Why is the topic important (or why do you study on it)? What are research questions? What has been studied? What are your contributions? The innovative and main contributions should be discussed in a more detailed way in light of a broader related literature and in terms of generalization of its findings.
  2. What is the actual and optimal composition of the samples?
  3. Author must explain the morphological structure and insights, which is missing in the SEM, TEM results.
  4. For photodegradation results: May refer to these literatures for mechanism discussion- New Journal of Chemistry, 2021, 2021,45, 9073-9083; Chemosphere, 2021, 287, 132225.
  5. A comparison table for the photocatalytic activity should be presented with the previous studies.
